# HandRefiner: Refining Malformed Hands in Generated Images by Diffusion-based Conditional Inpainting

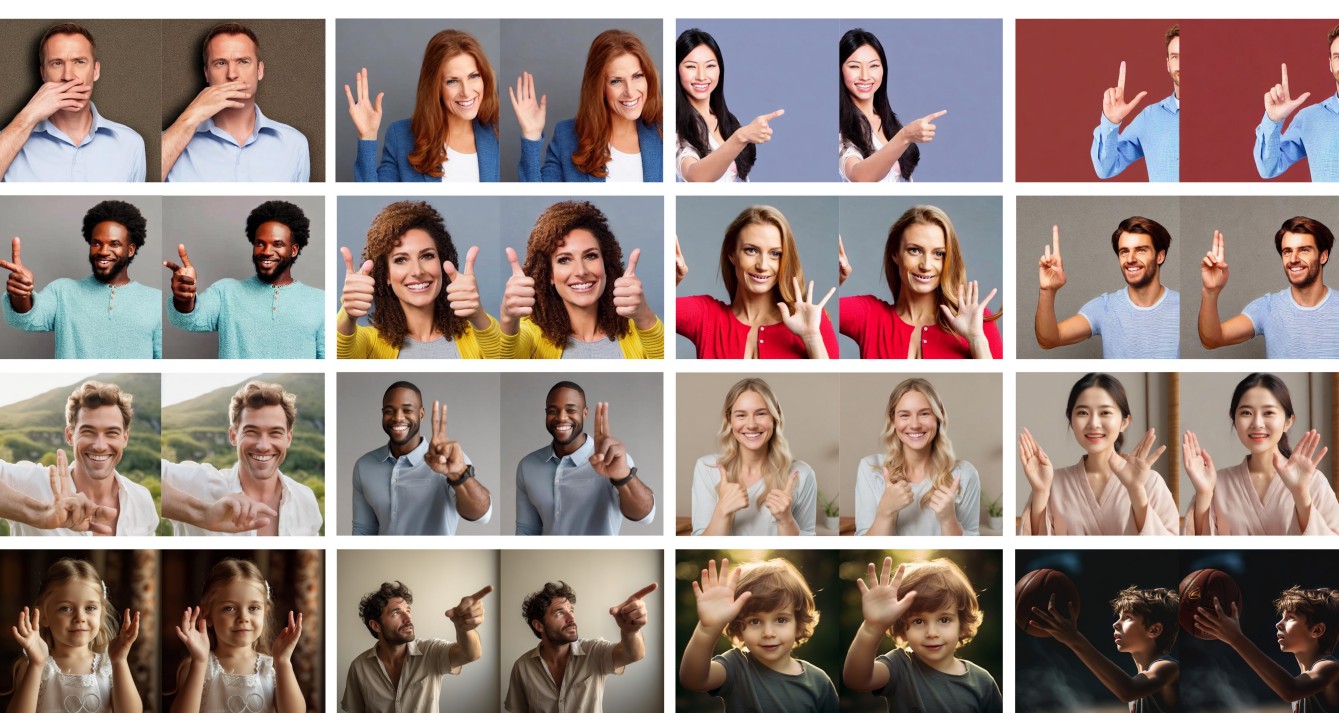

**Figure 1: Stable Diffusion [26] (first two rows), SDXL [21] (third row) and Midjourney [1] (last row) generate malformed hands (left in each pair), *e.g.*, incorrect number of fingers or irregular shapes, which can be effectively rectified by our HandRefiner (right in each pair).**

## ABSTRACT

Diffusion models have achieved remarkable success in generating realistic images but suffer from generating accurate human hands, such as incorrect finger counts or irregular shapes. This difficulty arises from the complex task of learning the physical structure and pose of hands from training images, which involves extensive deformations and occlusions. For correct hand generation, our paper introduces a lightweight post-processing solution called **HandRefiner**. HandRefiner employs a conditional inpainting approach to rectify malformed hands while leaving other parts of the image untouched. We leverage the hand mesh reconstruction model that consistently adheres to the correct number of fingers and hand shape, while also being capable of fitting the desired hand pose in the generated image. Given a generated failed image due to malformed hands, we utilize ControlNet modules to re-inject such correct hand information. Additionally, we uncover a phase transition phenomenon within ControlNet as we vary the control strength. It enables us to take advantage of more readily available synthetic data without suffering from the domain gap between realistic and synthetic hands. Experiments demonstrate that HandRefiner can significantly improve the generation quality quantitatively and qualitatively. The code will be released.

## CCS CONCEPTS

• **Computing methodologies → Reconstruction**; **Appearance and texture representations**; **Shape representations**; **Image processing**.

## KEYWORDS

Diffusion models, Deep learning, Image inpainting

*Conference acronym 'XX, June 03–05, 2018, Woodstock, NY*

© 2018 Copyright held by the owner/author(s). Publication rights licensed to ACM.

ACM ISBN 978-1-4503-XXXX-X/18/06

https://doi.org/XXXXXXX.XXXXXXX

# 1 INTRODUCTION

Diffusion-based models [9, 30, 32] have shown great success in text-to-image generation tasks [19, 21–24, 26, 28], *i.e.*, generating realistic images following text descriptions. These models, when trained on ample data, showcase a remarkable capacity for producing coherent structures corresponding to various descriptions and arranging them appropriately according to the textual cues.

Despite their impressive performance in generating a wide array of objects, diffusion-based models encounter challenges when it comes to generating realistic human hand images. As shown in Figure 1, for example, the women on the first row have incorrect numbers of fingers, while the men in the rightmost column have misshaped hands. We define such instances as "malformed", where a generated hand exhibits biological defects: incorrect anatomy or impossible poses, reflecting an intrinsic property of generated hands. This difficulty persists with both small-scale Stable Diffusion models [26] and large-scale Stable Diffusion XL models [21].

The primary reasons for this limitation can be attributed to two factors: (a) Inherently complex structure of human hands: A kinematic model of hand contains 16 joints and 27 degrees of freedom [7]. This indicates the pose space of a human hand is really complicated but also confined, generating a human hand has little tolerance for error. (b) Human hand is a 3D object: so various occlusions can occur between fingers when being projected to 2D image space. Depending on the hand pose and viewing perspective, one may observe varying numbers of fingers and different shapes.

Some methods [18] have attempted to enhance generation quality by introducing additional control signals like 2D human skeletons. However, the estimated 2D skeletons lack depth information, leading to ambiguity in defining precise hand structures. For instance, determining which finger is in the foreground can be challenging when fingers cross over each other.

To enhance the capability of existing methods in generating realistic human hands, we suggest incorporating precise human hand priors as guidance. Our approach involves utilizing a reconstructed hand mesh to provide essential information about hand shape and location. This information is then used to guide the generation of human hands within the diffusion models during inference. Moreover, rather than generating the entire image from scratch, we employ an inpainting pipeline to exclusively reconstruct the hand region while leaving the rest of the image untouched. This approach seamlessly integrates with existing diffusion models without necessitating re-training them.

The process begins with a generated image, we employ a hand mesh reconstruction model to generate the estimated depth map, incorporating essential hand pose and shape priors. This depth map then serves as guidance and is integrated into a diffusion model via ControlNet [38]. Besides, rather than relying on expensive paired real-world data for training, we identify a phase transition phenomenon within ControlNet when varying control strength. This discovery proves beneficial in ensuring the quality of generated hand shapes and textures when using synthetic data for training.

In summary, we make the following contributions. (1) We propose a novel inpainting pipeline HandRefiner to rectify malformed human hands in generated images. (2) We uncover a phase transition phenomenon within ControlNet, which is helpful in the usage of synthetic data for training. (3) The HandRefiner significantly enhances the quality of hand generation, as validated by comprehensive experiments showcasing improvements in both qualitative and quantitative measures.

# 2 RELATED WORK

## 2.1 Conditional Image Diffusion Model

By feeding into text descriptions as prompts, diffusion models have shown great success in generating realistic images following the semantics [6]. However, the text description alone is hard to express accurate information such as the pose of human bodies and the interaction gesture between different objects [35]. To provide more fine-grained conditions during the generation process, diverse control signals are explored in the prior studies. For example, ControlNet [38] and T2I-Adapter [17] utilize a set of extra image-level signals as inputs to control the generation results, including segmentation maps for layout control and 2D human skeletons for human pose guidance. To make the control signals more flexible, UniControl [39] feeds multiple control signals jointly into the diffusion models. Such an implementation allows the users to control different parts of the generated images using separate control signals jointly. Despite their good performance in generating realistic images, they do not always perform well in generating realistic humans, especially the human hands, due to the complex structure of human hands and the ambiguity of 2D skeletons. To further improve their generation quality on human hands, we introduce the HandRefiner in the paper, which leverages an inpainting pipeline built upon ControlNet to rectify the malformed hands in the generated images.

## 2.2 Generating Plausible Human Body Parts

Some prior works [29] have recognized the issue of misshaped hands generated by current text-to-image diffusion models. Many of these approaches [18] rely on user-provided information, such as precise human mesh data, to guide the image generation process. However, acquiring or preparing such data can be impractical during real-world applications. The closest work to ours [33] adopts a similar approach of using the output of a mesh reconstruction model to correct human whole-body poses. However, the task is considered very different due to the large differences in data availability, appearance, and relative size of the subject. Compared to the human hand, diffusion models have much better capability in generating plausible whole-body poses, primarily due to more training data and easier differentiability between arms, legs, and torso compared to fingers. Instead of using regeneration method in [33], our inpainting approach is better suited for hand rectification which precisely preserves the main character and context.

To assist existing methods in generating more realistic human hands, we propose a post-processing approach called HandRefiner, trained using synthetic data. Furthermore, our paper introduces a phase transition phenomenon, which helps bridge the gap between realistic and synthetic images. This method streamlines the data collection process and automates hand rectification without relying on additional inputs from users.

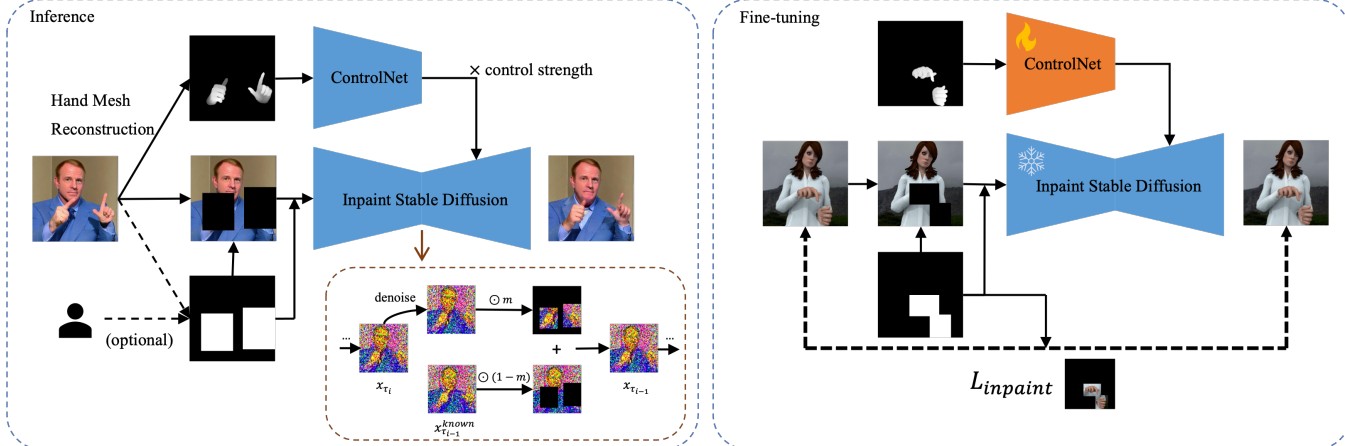

Figure 2: Overview of HandRefiner. During inference, the depth map fitted by a hand mesh reconstruction model is injected into the inpainting inference processes through ControlNet [38]. During fine-tuning, synthetic data is utilised to train the depth-based ControlNet to reconstruct hands following an inpainting pipeline. HandRefiner is simple yet effective, greatly enhancing the quality of hand generation.

## 3 METHOD

Figure 2 illustrates the overall pipeline of the proposed HandRefiner, including the inference pipeline (left part) and the training pipeline (right part), respectively. During inference, HandRefiner leverages a hand mesh reconstruction model to estimate the hand depth map from the given generated image. The model can capture the pose of the malformed hand, while also producing anatomically correct hand shape and structure. Then the hand regions are reconstructed via inpainting. During training, HandRefiner follows an inpainting pipeline, where the ControlNet receives the ground truth hand depth map as input and is fine-tuned using synthetic hand images while the Stable Diffusion model is frozen.

### 3.1 HandRefiner Pipeline

**Hand Reconstruction.** It aims to extract the hand mesh from a given generated image. Specifically, given an image containing malformed human hands, we first use mediapipe [37] to localize the hands automatically. Users can also optionally adjust estimated locations to get more accurate masks for the hand regions. Then, the state-of-the-art (SOTA) hand mesh reconstruction model, *i.e.*, Mesh Graphormer [13], is utilized to estimate regular hand mesh for each hand. This is attributed to that the model relies on learned patterns from the training data of normal hands and generalizes to generate plausible reconstruction even for malformed cases. However, the hand mesh information is hard to directly serve as a guide to existing Diffusion models due to its ambiguity in determining the hand's location and size in the 2D images. To this end, a depth renderer is further involved to render a depth map for each mesh.
**Inpainting.** Given the estimated depth map and the masks for hand regions, we first generate the masked images by masking the corresponding regions in the original image. After that, the masked image and the masks are treated as the inputs of the inpainting model, with the depth map as the control signal. We utilize the

inpainting Stable Diffusion model with ControlNet [38] as the inpainting model. Specifically, We feed the random initialized noise vector into the inpainting model and employ the DDIM sampler [31] to iteratively update the hand region. To ensure both the consistency of the non-hand region before and after the rectification process and the quality of the hands, we adopt a masking strategy similar to [16]. The original image is first projected to $x_0^{\text{known}}$. In each reverse step $i \in \{T, T-1, \ldots, 2\}$, the noise is added to projected feature $x_0^{\text{known}}$ to form $x_{\tau_{i-1}}^{\text{known}}$, *i.e.*,

$$x_{\tau_{i-1}}^{\text{known}} \sim N(\sqrt{\alpha_{\tau_{i-1}}} x_0^{\text{known}}, (1-\alpha_{\tau_{i-1}})\mathbf{I}), \tag{1}$$

where $N(\cdot)$ denotes the Gaussian distribution. $\alpha_{\tau_{i-1}}$ denotes a function of variance schedule defined in [31].

Then, we feed noise vector $x_{\tau_i}$ into the inpainting model to estimate the noise, and use DDIM sampler together with $x_{\tau_{i-1}}^{\text{known}}$ to obtain $x_{\tau_{i-1}}$, *i.e.*,

$$x_{\tau_{i-1}} = m \odot \text{DDIM}(\epsilon_\theta(x_{\tau_i}, x_{\text{mask}}, D)) + (1-m) \odot x_{\tau_{i-1}}^{\text{known}}, \tag{2}$$

where $m$ is the downsampled mask of the mask generated in the hand reconstruction stage with '1' corresponding to the hand regions and '0' for other regions. $x_{\text{mask}}$ comprises the mask $m$ and the projected masked image. They are concatenated with the noise vector $x_{\tau_i}$ as the input to the inpainting Stable Diffusion model in each iteration. $D$ is the converted depth map according to the reconstructed hand mesh to inject hand shape prior to the diffusion process. DDIM$(\cdot)$ and $\epsilon_\theta(\cdot)$ denote a DDIM sampling step and the inference process of the inpainting model, respectively. $\odot$ is the Hadamard product. In the last step, a single DDIM step without masking is used to ensure harmony between hand and non-hand regions in denoised features. The denoised features are then sent into the decoder to re-generate the images with rectified human hands. Note only the parameters of the $\epsilon_\theta$ model are tunable during the training process.

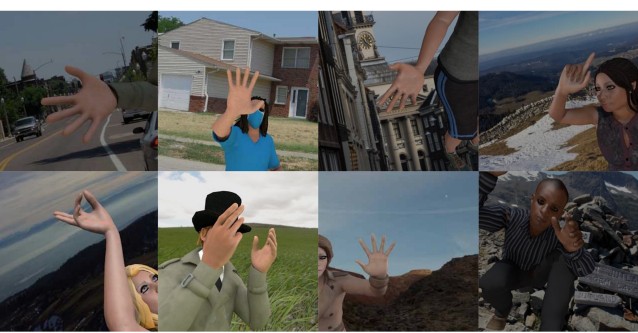

**Figure 3: Hands generated by using the pre-trained Depth ControlNet in our HandRefiner pipeline.**

**Figure 4: Examples of hands in the RHD dataset [40], which lack real textures.**

## 3.2 Phase Transition

Given the proposed HandRefiner, one naive solution is to apply pre-trained depth ControlNet in the pipeline. However, we found it is insufficient to produce satisfactory results. The original depth ControlNet was trained with general images with paired whole-image depth maps estimated by depth estimation models [4, 25, 36]. When conditioned on a depth map rendered from the reconstructed hand mesh, it may not recognize the hand, thereby introducing extra artifacts, or creating an inharmonious connection between hands and wrists, as shown in Figure 3. Hence, further fine-tuning is needed to improve the hand generation quality.

It is rather hard to collect the paired real-world data with accurate depth maps and corresponding human images. Using synthetic data for training can greatly alleviate such problems while introducing bias in the generated images. As shown in Figure 4, the images in the synthetic datasets may contain much fewer wrinkles than the realistic images. Such a domain gap will mislead the diffusion model, leading to unrealistic images lacking real textures.

Fortunately, during the training of ControlNet with synthetic images, we discovered an interesting phase transition phenomenon in the generation process. The base stable diffusion network, initially trained on realistic data, exhibits the capability to generate realistic images. By adjusting the control branch strength, we can strike a balance between the network's ability to produce realistic images and precisely adhere to guidance at the cost of realism. This involves linearly scaling the output of each ControlNet encoder block

by a designated control strength and then integrating the encoded feature into the stable diffusion model. As depicted in Figure 5, the control signal predominantly influences the structure-following ability in generated images when the control strength is below 0.5. Higher control strength makes the generated hands more inclined to follow the structure outlined in the control signal, such as the depth map used in our HandRefiner. However, excessive control strength leads to texture degradation, resulting in hands with significantly reduced texture. Hence, finding the optimal 'sweet spot' for control strength to achieve both correct structure and realistic texture, is crucial for leveraging synthetic images for training. We present two strategies: a fixed control strength strategy and an adaptive control strength strategy.

**Fixed Control Strength.** We first collect a set of images containing malformed hands and randomly sample 200 images from them. Then, we use the proposed HandRefiner pipeline to rectify the generated hands with a series of predefined thresholds of control strength. After that, we manually determine the best threshold for each image based on two factors: hand anatomy and texture. Finally, we determine the fixed control strength by taking the average. As shown in Figure 6, since the distribution is tightly centered, the average value should be a close estimate of the human-preferred control level. This simple strategy with fixed control strength achieves a great balance between speed and generation quality.

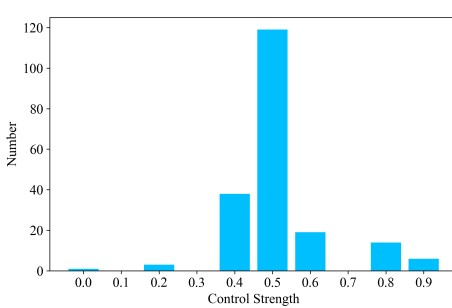

**Figure 6: Distribution of selected control strength over 200 images.**

**Adaptive Control Strength.** Although the simple fixed control strength can facilitate the hand generation in both structure following and texture keeping in most cases, it may fail in extreme cases. To this end, we also propose a simple adaptive control strength strategy, which requires iterating over only a few predefined thresholds and generating the images accordingly. The quality of these images is then evaluated by employing 2D MPJPE (Mean Per Joint Position Error) [10] as a metric, with the original hand mesh serving as ground truth. Specifically, we first apply the same hand mesh reconstruction model to obtain the hand mesh of the generated hands in each image. Then, a sparse linear regressor [27] is utilized to estimate the 3D keypoint locations from the mesh vertices. Next, we project the coordinates to 2D image space using the same pinhole camera model we used to render the depth map, so the error is in unit pixel. The optimal threshold is determined as the minimum strength where MPJPE falls below a fractional multiple of the reference error, *i.e.*, the error when control strength is set to 1. The process can be summarised as pseudo-code:

Figure 5: Illustration of Phase Transition. The hand shape and pose change to match the depth map when applied control strength is small, while the hand wrinkles and textures disappear at high control strength.

```
1:  procedure ADAPTIVE THRESHOLD(groundtruth)
2:      selected = sample(1.0)
3:      ref_error = MPJPE(groundtruth, detect(selected))
4:      for strength ← 0.4 to 0.9 do
5:          x = sample(strength)
6:          error= MPJPE(groundtruth, detect(x))
7:          if error < ref_error × 1.15 then
8:              selected = x
9:              break
10:     return selected
```

## 3.3 Negative Prompting

To further enhance the quality of the generated images, we manually append negative prompts during the generation process. Here, we involve one simple negative prompt $c_{n0}$ (*i.e.*, 'fake 3D rendered image') during inference since it is not this paper's focus and more negative prompts can be explored in the future. Formally, given the positive prompt $c_p$ and the standard negative prompt $c_n$ (*e.g.*, 'long body', 'low-resolution', etc.), we form the score input to each DDIM sampling step by:

$$\tilde{\epsilon}_\theta = \epsilon_\theta(x_t, c_{n0} + c_n) + w(\epsilon_\theta(x_t, c_p) - \epsilon_\theta(x_t, c_{n0} + c_n)), \quad (3)$$

where $w$ denotes the guidance strength.

## 3.4 Loss Function

We only tune the control branch during training. Instead of using the whole image as the reconstruction target, we encourage the model to focus on the hand regions by using the hand region mask $m$ in the following denoising loss:

$$L = \mathbb{E}_{t,x_0,\epsilon}\left[\left\|\frac{m}{\sum m} \odot \left(\epsilon - \epsilon_\theta(\sqrt{\alpha_t}x_0 + \sqrt{1-\alpha_t}\epsilon, t)\right)\right\|^2\right]. \quad (4)$$

## 4 EXPERIMENTS

### 4.1 Experiment Settings

**Datasets.** We use a combination of RHD [40] and Static Gesture Dataset [3] to fine-tune the proposed HandRefiner. During training, 3D-rendered third-person view RGB images containing diverse backgrounds, along with corresponding depth maps and segmentation maps from both datasets are utilized. The segmentation maps are utilized to generate the depth and mask for the hand regions. BLIP [12] is employed for caption generation on each image from these two datasets. We adopt the FreiHAND [41] and HAGRID [11] datasets for evaluation to ensure the high quality and variety of the test data. We calculate the frequency of each gesture in the HAGRID dataset to formulate the prompts used in the evaluation. **RHD** comprises 41,258 images from 16 subjects covering 31 actions, which are resized to 512×512 during training. **Static Gesture Dataset** is a smaller dataset with 10k images from 100 different subjects. Approximately 3k images with incorrect segmentation masks are filtered out. We randomly crop images around the hand region with a size of 512×512 to generate the training samples. **HAGRID** is a larger dataset and offers 0.55M high-resolution RGB images captured mainly indoors, spanning 18 common daily-life gesture categories. Each image contains a person with different hand gestures. **FreiHAND** is a standard benchmark for hand mesh reconstruction, including 32,560 samples with diverse hand gestures. Each sample has four views centering on a hand.

**Metrics.** We use both objective and subjective evaluation to thoroughly demonstrate the effectiveness of the proposed HandRefiner model. Specifically, we evaluate the quality of generated images by using standard metrics: Frechet Inception Distance (FID) [8] and Kernel Inception Distance (KID) [5]. We also use the keypoint detection confidence scores of a hand detector [15, 37] to indicate the plausibility of generated hands. Furthermore, we adopt the 2D MPJPE [10] to evaluate the pose reconstruction error of generated

| Method | FID ↓ | KID ↓ | Det. Conf. ↑ | Gesture Prompt |
|--------|-------|-------|--------------|----------------|
| Stable Diffusion | 77.60 | 0.074 | 0.934 | Yes |
| **HandRefiner** | 74.12 | 0.071 | 0.944 | Yes |
| Stable Diffusion | 85.71 | 0.083 | 0.936 | No |
| **HandRefiner** | 81.80 | 0.079 | 0.945 | No |

Table 1: Comparison of image-level FID and KID between original images and rectified images using HAGRID as the reference.

hands. Additionally, to gather subjective evaluations from a human perspective, we conduct user surveys on the generated images.

**Implementation Details.** We employ the Stable Diffusion v1.5 [26] fine-tuned inpainting model as our base model, maintaining its parameters frozen. The ControlNet branch is initialized with weights from the depth-guided version, as we leverage the depth map for guidance during generation. The ControlNet branch is further fine-tuned with synthetic data for 2,307 steps, using the AdamW optimizer [14] with a batch size of 16. A fixed learning rate of 2e-5 is applied during training. The training and testing of models are conducted on A100 GPUs, with a default control strength set to 0.55 in our experiments.

## 4.2 Comparison with Stable Diffusion Baseline

**HAGRID as the Reference.** We generate 12K images by first sampling the text descriptions from the HAGRID dataset, *i.e.*, covering 18 diverse gestures, and then feeding the descriptions into the Stable Diffusion model. The generated images are then processed by the proposed HandRefiner. The results are summarized in the first two rows in Table 1, which is denoted by using 'Gesture Prompt'. It can be observed that although the hand region occupies a small region in the whole image, our HandRefiner can greatly improve the FID metric by more than 3 points, *i.e.*, from 77.60 to 74.12. It can also improve the keypoint detection confidence by 1 point, *i.e.*, from 93.4% to 94.4%.

Apart from evaluating the models' ability based on the most common 18 gestures, we also benchmark the proposed HandRefiner's performance on arbitrary gestures by feeding the following relaxed prompt *'a person facing the camera, making a hand gesture, indoor'* for image generation. As shown in the last two rows in Table 1, HandRefiner can improve the hand generation quality in such a case by about 4 FID, *i.e.*, from 85.71 to 81.80. Such observation indicates that our HandRefiner can greatly improve the hand generation quality in both restricted and relaxed constraints.

**FreiHAND as the Reference.** Different from the HAGRID dataset which contains both human and human hands, the FreiHand dataset only contains hands and can provide a more accurate evaluation of the hand generation quality. We sample 12K images using the stable diffusion model. We carefully filter out the images where the generated hands are too small or with intersecting hands to follow the FreiHand distribution. The images are then processed by HandRefiner to rectify the malformed hands. After the generation, we simply crop the hands from the generated images and use the cropped images for evaluation, formulating a total of 15K cropped

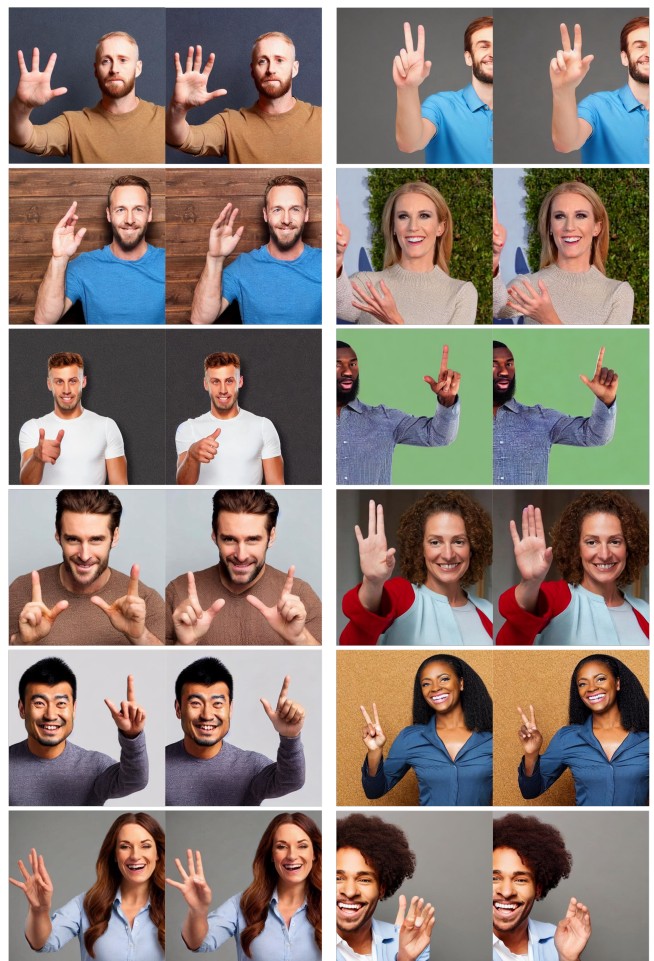

Figure 7: Visual results on the HAGRID dataset.

| Method | FID ↓ | KID ↓ | Det. Conf. ↑ |
|--------|-------|-------|--------------|
| Stable Diffusion | 158.34 | 0.117 | 0.936 |
| **HandRefiner** | 147.52 | 0.107 | 0.948 |

Table 2: Comparison of hand-level FID and KID between original images and rectified images using FreiHAND as the reference.

images. As shown in Table 2, HandRefiner greatly improves Stable Diffusion's hand generation quality by over 10 FID. It further demonstrates that the proposed HandRefiner can improve the hand generation quality in both human-hand scenarios and hand-only scenarios, further validating its effectiveness.

**User Study.** Apart from the objective evaluation, we also conduct the subjective test with 50 participants. Specifically, we sample 100 images randomly generated from the Stable Diffusion model and rectify them using the HandRefiner. We then put each pair of original and rectified images side-by-side to formulate the subjective survey. Each participant is asked to select the image that has

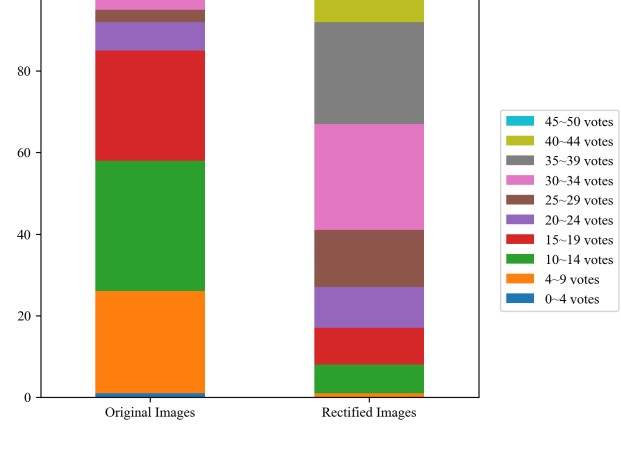

**Figure 8: Human Evaluation Results. The color represents the number of positive votes from 50 participants.**

more real human hand(s). We also provided a 'neither' option to prevent inflation in survey responses. The survey is conducted on the Amazon Mechanical Turk platform to ensure professionalism and fairness. Here we intentionally do not provide the definition of 'real' to users to fully expose users' subjective opinions. Figure 8 visualizes the results. It shows the rectified images significantly outperform the original images, *i.e.*, 70 out of 100 rectified images have the majority of votes (>50%), and 84 were preferred over other options. Such phenomena indicate HandRefiner can greatly improve hand generation quality in terms of human visual experience.

Among the 100 depth maps reconstructed from the malformed survey images covering diverse indoor and outdoor scenarios, 97 capture the correct topology of standard hands, *i.e.*, a palm with five fingers. Only 9 depth maps have a slight unnatural distortion in finger(s). This demonstrates the reliability and robustness of the Mesh Graphormer, as it can accommodate the majority of malformed inputs effectively. Mesh Graphormer is trained on a closed set with correct hands, thus having a strong regularization to generate accurate hand representations.

## 4.3 Ablation Studies

To distinguish the effectiveness of each design adopted in the proposed HandRefiner, we utilize a subset design, *i.e.*, sampling 2,000 images following the gesture distributions in the HAGRID dataset. The hand regions are then masked in these images and the HandRefiner is utilized to re-generate the hands under different configurations. We utilize both quantitative and qualitative comparisons to demonstrate the effectiveness of our design.
**Effect of Fine-tuning.** We first investigate the effectiveness of fine-tuning with synthetic hand data. The ControlNet that utilizes the general depth map as the control signal during training is treated as the baseline method. It can be observed from the first two rows of Table 3 that the utilization of paired hand data and depth maps which are generated from hand meshes can greatly improve the hand generation quality, *i.e.*, from 18.971 FID to 14.357 FID and

17.722 MPJPE to 8.576 MPJPE. As shown in Figure 9 (a), the ControlNet without hand data fine-tuning can not generate the hand structure well, *e.g.*, the generated hands have extra fingers. The generated hands after the HandRefiner rectification have more reasonable structures, indicating that the introduced paired hand data can aid the model more focus on the details of the hand structures and better utilize the depth information during hand generation.

| Fine-tuned | Control Strength | Negative Prompt | Inpainting Loss | FID ↓ | MPJPE ↓ |
|---|---|---|---|---|---|
| ✗ | 0.55 | ✗ | | 18.971 | 17.722 |
| ✓ | 0.55 | ✗ | ✓ | 14.357 | 8.576 |
| ✓ | 0.55 | ✓ | ✗ | 13.989 | 8.078 |
| ✓ | 0.55 | ✓ | ✓ | 13.977 | 7.878 |
| ✓ | 1.0 | ✓ | ✓ | 14.959 | 7.150 |
| ✓ | adaptive | ✓ | ✓ | 13.823 | 6.330 |

**Table 3: Ablation study of the design choice of HandRefiner.**

**Effect of Negative Prompt.** We further explore the usage of negative prompts during the generation process. Note that the default negative prompts in the generation process remain unchanged and we only investigate the influence of the added negative prompt, *e.g.*, "fake 3D rendered image". As shown in the $2^{nd}$ and $4^{th}$ rows of Table 3, the involvement of such a simple negative prompt brings a better performance in both FID and pose error, *e.g.*, 0.380 FID and 0.698 MPJPE improvement. It should be noted that it is almost free to add such negative prompts during inference as it does not change the training process. It can be observed in Figure 9 (b) that involving such negative prompts can encourage the model to generate more rich textures in the hands, resulting in better generative quality.
**Effect of Inpainting Loss.** We also utilize different loss functions to train the HandRefiner model, including the naive diffusion loss and the modified inpainting loss, *i.e.*, only calculating the loss in the hand regions. As shown in the $3^{rd}$ and $4^{th}$ rows of Table 3, using the inpainting loss can guide the model to more focus on the hand regions, leading to reduced pose errors with no extra computational costs during inference.
**Effect of Control Strength.** As demonstrated in the phase transition section, control strength is an effective factor in determining the hand generation quality. We compare the performance of HandRefiner with the fixed optimal control strength of 0.55, the fixed default control strength of 1.0 as in other works, and the adaptive control strength strategy. The results are shown in the last three rows of Table 3 and Figure 9 (c). It can be summarized that using a high fixed control strength like 1.0, the model has a lower pose error, but a higher FID score, indicating that the higher control strength will make the model better follow the structure but lacks detailed textures since the synthetic data are used during training. The adaptive control strength strategy further improves the visual and quantitative results, while at the cost of more than 3.5 times inference costs. To this end, we tend to utilize the fixed optimal control strength by default and leave the adaptive control strength strategy as an optional solution.

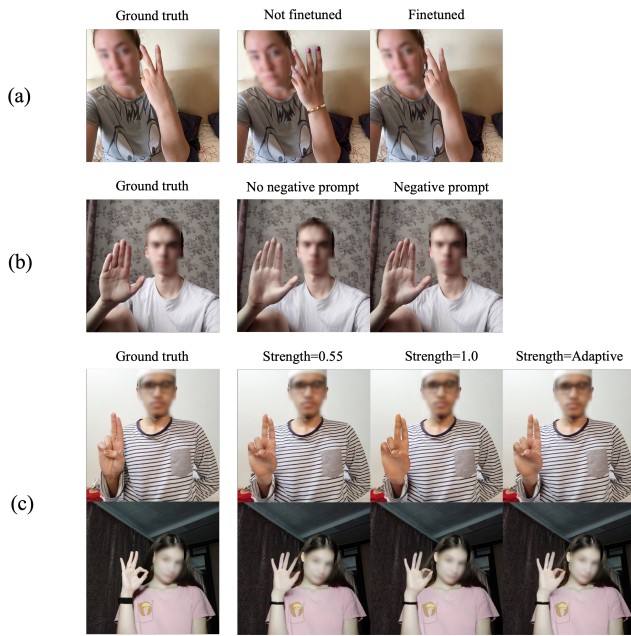

Figure 9: Comparison of different methods on repainting 2k HAGIRD images.

## 4.4 Further Exploration on Phase Transition

It is also interesting to demonstrate that the phase transition phenomena are not isolated in hand generation tasks, but also can be observed in other settings. Specifically, we train ControlNet using other kinds of control signals on corresponding synthetic datasets, *e.g.*, we use the FaceSynthetics [34] dataset to train two ControlNets conditioned on face landmarks and segmentation masks, respectively. Then we vary control strength during inference, and the same phase transition phenomenon is observed, *e.g.*, the person's pose is mainly influenced when the control strength is small, while larger control strength mainly changes the style and texture, as shown in Figure 10. A similar conclusion can be drawn from the ControlNet conditioned on the surface normal map of synthetic people provided by Static gesture dataset [3] and Animated gesture dataset [2]. We can apply this technique to other domains like animal image generation with accurate pose control.

## 5 LIMITATIONS AND DISCUSSION

The proposed HandRefiner can greatly improve the hand generation quality in current stable diffusion works. The post-processing nature of the HandRefiner makes it suitable for different generation-based methods. Although we conduct extensive experiments in the paper and demonstrate our effectiveness, more kinds of models can be explored, *e.g.*, DiT models [20]. Besides, although we focus on free-form hand generation as current diffusion models already face great challenges when generating free hand images, HandRefiner can generalize to interacting hands as shown in Figure 11. Due to the limited training data and challenging mesh reconstruction under heavy occlusions, it may face difficulty in generating more complex scenarios, *i.e.*, intertwined fingers. It will be our future

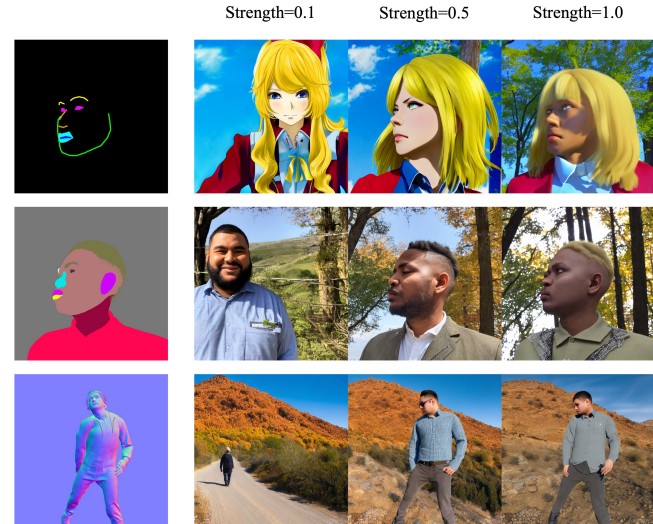

Figure 10: Phase transitions in ControlNet conditioned on different control signals. All models are trained with only synthetic data.

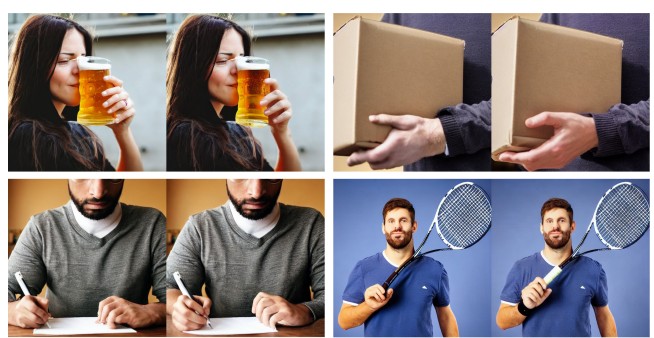

Figure 11: Simple hand-object interaction examples that are rectified by the HandRefiner.

work to target such cases and further improve the proposed method. Furthermore, we mostly focus on the generation quality of hands with appropriate sizes, *i.e.*, dismissing the cases with extremely small hand regions. Such small hands can be fixed by applying super-resolution to the original image or using a higher resolution base model like SDXL [21].

## 6 CONCLUSION

We present a novel conditional inpainting method based on ControlNet to rectify malformed human hands in generated images. By discovering an interesting phase transition phenomenon when varying control strength, we allow model training on easily available synthetic data, while still maintaining realistic generation results. We propose two simple techniques to determine the control strength in inference, and quantitatively and qualitatively verify their effectiveness. Lastly, we demonstrate the phase transition is generalizable to other control signals and settings.

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
