# OpenReview forum: "HandRefiner: Refining Malformed Hands in Generated Images by Diffusion-based Conditional Inpainting"
_acmmm.org/ACMMM/2024/Conference — MM2024 Poster_

### Official Review · Reviewer_6PVm · 2024-05-24

**Rating:** 4
**Confidence:** 2

**Summary:**

The paper introduces HandRefiner, a post-processing tool designed to correct malformed human hands in images generated by diffusion models.

**Strengths:**

The use of a hand mesh reconstruction model to provide accurate anatomical details is a strong technical contribution. This ensures that the generated hands adhere to realistic shapes and poses.
The integration of ControlNet for injecting hand shape information during the inpainting process is technically sound and enhances the generation process.

**Limitations:**

1.The issue discussed in this article is limited in scope. Although it is a real problem, it seems that such methods have not been published as academic papers.
2.More experiments are needed. The comparions are limited to Stable Diffusion. Why not to compare more Image Generation models to show your approach's ability.
After careful consideration, I have decided to give a borderline accept. This is because the method does indeed demonstrate the ability to solve a practical problem. However, as an academic article, I believe it requires further research and discussion.

**Suitability:**

2

---

### Official Review · Reviewer_t4Fj · 2024-05-24

**Rating:** 5
**Confidence:** 2

**Summary:**

The paper proposes to correct hand regions from a generated image using stable diffusion. Instead of generating the whole image again based on given conditions from scratch, the paper formulates the problem as an inpainting problem and only inpaints the wrong hand regions. It proposes to use rendered depth map as the guidance and achieve impressive results in generating images with faithful hands.

**Strengths:**

- It proposes a novel inpainting framework to correct wrong hand regions in images generated by stable diffusion.
- It introduces to use the rendered depth map as the condition for generation.
- The control strength strategy proposed by the work achieves a good balance between plausible hands and texture reality.
- It achieves good performance when compared with the stable diffusion baseline.

**Limitations:**

- Missing ablation experiments about control signals. The paper proposes to use rendered depth to control the generation, but has not quantitatively compare with other conditions (e.g., normal maps, segmentation maps, contours). Also, controlnet allows to take multiple conditions as inputs. Why not take a combination of these conditions to generate more faithful results?
- Missing to compare with other inpainting methods. Actually, there are many inpainting methods (e.g., [a]). The author chooses to build inpainting network based on controlnet. To make a comparison with other inpainting methods could show the advantage of the proposed method in an more obvious way.

[a] MAT: Mask-Aware Transformer for Large Hole Image Inpainting

**Suitability:**

2

---

### Official Review · Reviewer_N8uY · 2024-05-24

**Rating:** 3
**Confidence:** 4

**Summary:**

The authors utilize ControlNet and Inpaint Stable Diffusion to rectify the malformed hands in generated images. Specifically, given a malformed hand image, they first utilize existing hand tracking and hand-mesh reconstruction models to obtain mask and hand depth map, and input the mask and masked images to the inpainting model, which is conditioned on the depth map. They also finetuned the model with synthetic hand data to improve the performance. Both quantitative and qualitative results show the effectiveness of their model.

**Strengths:**

1.	Both the quantitative and qualitative results are impressive.
2.	The phase transition seems an interesting phenomenon, which might motivate other related fields.

**Limitations:**

1.	The paper directly applies the ControlNet and masked-based Stable Diffusion on refining malformed hands, which seems not that novel.
2.	In line 319, The author mentioned that they adopt a masking strategy similar to [16]. But In all the experiments they compare the result of HandRefiner with that from Stable Diffusion. I'm wondering about the result of directly adopting the method in [16] on this task.
3.	Could the author provide the qualitative result of using the FreiHAND as the reference?
4.	In line 7 of the pseudo-code, how do the authors come up with the multiplier of 1.15?

**Suitability:**

3

---

### Official Review · Reviewer_4xeE · 2024-06-03

**Rating:** 4
**Confidence:** 3

**Summary:**

This paper introduces HandRefiner, a post-processing solution for refining malformed hands in generated images. ​ The authors address the challenge of generating accurate human hands by leveraging a hand mesh reconstruction model and a conditional inpainting approach. ​ The HandRefiner pipeline involves estimating the hand depth map, inpainting the hand regions, and integrating the depth map into the diffusion model. ​ The key methods include hand mesh reconstruction using mediapipe and a hand mesh reconstruction model, conditional inpainting for hand region reconstruction, training of ControlNet using synthetic data, the use of negative prompting and a loss function focusing on hand regions.
The results show that HandRefiner significantly improves hand generation quality, as demonstrated by reduced FID and MPJPE scores compared to the baseline, and is effective in various gesture constraints. ​ User surveys confirm the subjective improvement in hand generation quality, and ablation studies highlight the effectiveness of various components of the approach.

**Strengths:**

Strengths of this paper are as follows:
1.
2. Theoretical approach is well-defined and supported by established models and methods. It involves leveraging a hand mesh reconstruction model to provide essential information about hand shape and location. ​ This information is then used to guide the generation of human hands within diffusion models during inference. ​The approach also incorporates conditional inpainting to exclusively reconstruct the hand region while leaving the rest of the image untouched.
3. Clarity:The paper is easy-to understand because of its well-organized structure, and visual aids.

**Limitations:**

I have a few comments on the limitations of this paper:

1. Novelty: While the proposed HandRefiner approach is effective in refining malformed hands in generated images, it builds upon existing techniques such as hand mesh reconstruction and conditional inpainting. ​ Therefore, the novelty of the paper lies more in the combination and application of these techniques rather than introducing entirely new concepts.
The use of ControlNet modules and the observation of a phase transition phenomenon are interesting findings, but additional explanation would be helpful in determining the contribution in terms of novelty.

2. Evaluation & Application: It would be interesting to know the author's perspective on how does fixing the malformed hands as a post-processing step compare to directly handling this at the generation step. More specifically, in context with a similar work that was accepted to CVPR 2024:

Supreeth Narasimhaswamy, Uttaran Bhattacharya, Xiang Chen, Ishita Dasgupta, Saayan Mitra, Minh Hoai; HanDiffuser: Text-to-Image Generation With Realistic Hand Appearances, https://doi.org/10.48550/arXiv.2403.01693

**Suitability:**

3

---

### Meta-Review · Area_Chair_kJ9P · 2024-07-01

**Recommendation:** Accept (Poster)
**Confidence:** 5

**Metareview:**

The paper introduces HandRefiner, a post-processing solution designed to refine malformed hands in generated images. The approach leverages a hand mesh reconstruction model and a conditional inpainting method to address the challenges of accurately generating human hands. The pipeline involves estimating a hand depth map, inpainting hand regions, and integrating this map into a diffusion model, using mediapipe for hand mesh reconstruction and conditional inpainting for specific region reconstruction. Results, including reduced FID and MPJPE scores, demonstrate significant improvements in hand generation quality, supported by user surveys and ablation studies.

Strengths:

+ The paper employs established models and methods, such as hand mesh reconstruction and conditional inpainting, to effectively guide the generation of human hands.

+ The structure of the paper and the use of visual aids contribute to its clarity and ease of understanding.

Limitations:

+ The novelty of the paper is more about the application and combination of existing techniques rather than the introduction of new methodologies.

+ There is a need for a broader comparison, particularly with methods that handle malformed hands during the initial generation rather than post-processing.

Overall, the paper is well-positioned within multimedia processing due to its innovative approach to a specific problem, though it could benefit from a deeper exploration of its novel contributions and comparative analysis with other contemporary methods. The recommendation leans towards a borderline acceptance.